# Study on Air Quality and Its Annual Fluctuation in China Based on Cluster Analysis

**DOI:** 10.3390/ijerph19084524

**Published:** 2022-04-08

**Authors:** Shengyong Zhang, Yunhao Chen, Yudong Li, Xing Yi, Jiansheng Wu

**Affiliations:** 1Key Laboratory for Urban Habitat Environmental Science and Technology, School of Urban Planning and Design, Peking University, Shenzhen 518055, China; zhangshengyong@stu.pku.edu.cn; 2School of Urban Planning and Design, Peking University, Shenzhen 518055, China; chenyunhao@stu.pku.edu.cn (Y.C.); liyd@stu.pku.edu.cn (Y.L.); yx2020@stu.pku.edu.cn (X.Y.); 3Key Laboratory for Earth Surface Processes, Ministry of Education, College of Urban and Environmental Sciences, Peking University, Beijing 100871, China

**Keywords:** air quality and fluctuation, clustering analysis, kriging interpolation, principal component analysis

## Abstract

Exploring the spatial and temporal distribution characteristics of air quality has become an important topic for the harmonious development of human and nature. Based on the hourly data of CO, O_3_, NO_2_, SO_2_, PM_2.5_ and PM_10_ of 1427 air quality monitoring stations in China in 2016, this paper calculated the annual mean and annual standard deviation of six air quality indicators at each station to obtain 12 variables. Self-Organizing Maps (SOM) and K-means clustering algorithms were carried out based on MATLAB and SPSS Statistics, respectively. Kriging interpolation was used to get the clustering distribution of air quality and fluctuation in China, and Principal Component Analysis (PCA) was used to analyze the main factors affecting the clustering results. The results show that: (1) Most areas in China are low-value regions, while the high-value region is the smallest and more concentrated. Air quality in northern China is worse, and the annual fluctuations of the indicators are more dramatic. (2) Compared with AQI, AQFI has a strong indication significance for the comprehensive situation of air quality and its fluctuation. (3) The spatial distribution of SOM clustering results is more discriminative, while K-means clustering results have a large proportion of low-mean regions. (4) PM_2.5_, PM_10_ and CO are the main pollutants affecting air quality and fluctuation, followed by SO_2_, NO_2_ and O_3_.

## 1. Introduction

As the basic condition for the survival of all living things on earth, the atmosphere is the basis of urban production activities, and its quality directly affects the balance of the whole ecosystem and the survival and development of human society [1]. However, with the rapid economic and social development, the acceleration of urbanization, the adjustment of industrial structure, the increase of energy consumption, population and the number of motor vehicles, people are faced with a series of air pollution problems caused by unbalanced development. Therefore, air quality issues are getting more and more attention [2]. According to statistics, the harm of air pollution to human health has become one of the obstacles affecting people’s quality of life, and air quality has risen from an actual livelihood issue to a national development issue. Under such a severe background, it has become an important topic for the harmonious development of man and nature to explore the spatio-temporal distribution characteristics of air quality and seek pollution control measures [3].

There are many research results on air quality at home and abroad, mainly focusing on the following three aspects: (1) Explore the spatial-temporal distribution characteristics of air quality at different scales and with different methods [4,5,6,7]. (2) Study the influencing factors of air quality [4,8,9,10,11,12]. (3) Research on the air quality prediction model and improvement of the calculation method [13,14,15,16,17]. However, few studies have focused on the annual fluctuation of air quality and its spatial distribution, which can reflect the livable degree of air in a region, and its spatial distribution has a guiding significance for air governance in different regions. 

In order to better explore air quality, studies always use some pollutant concentrations as indicators of air pollution status. Wang et al. [18] employed the hourly average concentration data on the air quality index (AQI) and six component pollutants (PM_2.5_, PM_10_, SO_2_, NO_2_, CO and O_3_) in 336 Chinese cities from 2014 to 2019. Nguyen et al. [19] utilized multi-platform datasets to elucidate two trans-boundary PM_2.5_ episodes in HCMC, Vietnam, over the periods 25–29 October 2013 and 05–08 October 2015. The dataset chosen for their study includes six criteria air pollutants (i.e., PM_2.5_, PM_10_, SO_2_, NO_2_, CO and O_3_) for the entire China region. Mahato et al. [20] divided the study period into four phases: the first and second cycles before the first-level prevention confinement and the first and second periods of the first-level prevention confinement, and used the average air quality index (NAQI) to explore the difference in the impact of the blockade on air quality in different stages. The NAQI data cover SO_2_, CO, NO_2_, O_3_, NH_3_, PM_2.5_ and PM_10_. As an important indicator of air quality standards, the AQI (Air Quality Index) monitors six pollutants: sulfur dioxide, nitrogen dioxide, PM_10_, PM_2.5_, carbon monoxide and ozone, and the data is updated every hour. AQI presents these six pollutants with a unified evaluation standard [18]. So, these six pollutants are widely used indicators in the field of air quality. Combined with this paper’s exploration of the meaning of these pollutants and their impact on air quality, in this research, we finally select these six pollutants as indicators to characterize air quality and its fluctuations. These six indicators all affect the regional air quality to varying degrees, and the specific indicators include CO [21], O_3_ [22], NO_2_ [23], SO_2_ [24], PM_10_ [23] and PM_2.5_ [25]. The meanings of them are shown in Table 1.

This research is based on the hourly monitoring data of six air quality indicators at 1427 stations in mainland China in 2016. A total of 72 days with dates ending in 1 and 6 were screened out in each month (except 31). The hourly average data of filtered dates were selected to calculate the annual mean and annual standard deviation of six air quality indicators at each station, and 12 variables were obtained, which were used to represent the comprehensive Air Quality and Fluctuation Index (AQFI). After Z normalization of the variables, we delete the site data with null data. MATLAB, SPSS (program Statistical Package for the Social Sciences) and other software were used to make SOM and K-means clustering analysis on the variables, and kriging interpolation was used to get the clustering distribution of air quality and fluctuation in China. Finally, principal component analysis was used to analyze the main factors affecting the clustering results. The technology roadmap is shown in Figure 1.

## 2. Materials and Methods

### 2.1. Study Area and Data Sources

The research area of this paper is China, and the hourly monitoring data of six pollutant indicators of 1427 stations in China in 2016 are obtained. The national air quality data comes from the national urban air quality real-time release platform of the China Environmental Monitoring Station, which is updated daily, and Beijing air quality data from the Beijing Environmental Protection Testing Center website, which is updated daily. The distribution of meteorological stations is shown in Figure 2.

### 2.2. Methods

#### 2.2.1. Clustering Analysis Based on MATLAB—Self-Organizing Maps (SOM)

The learning algorithm of SOM is similar to that of competitive neural networks, and both adopt Kohohen learning rules [26]. The main difference between the two is that there is no mutual connection between core layers in the competitive neural network, and only the corresponding connection weight of the winning neuron is updated in the winner-takes-all way. In the self-organizing mapping network, the neurons in a certain area near each neuron will also be updated, but the distant neurons will not be updated, so that the geometrically similar neurons become more similar [27].

The training steps of the SOM neural network are as follows: first, set the weight vector between the sample and the *i*th input node and output neuron, initialize the weight with a small random value, and normalize the input vector and weight. A random sample is then fed into the network. The sample is the inner product with the weight vector, and the output neuron with the largest inner product wins the competition. Since the sample vectors and weights are normalized, maximizing the inner product is equivalent to minimizing the Euclidean distance. For the neurons in the field of winning neuron topology, Kohonen rules are used to update, and different distance functions can be used to determine the field, such as Euclidian distance (Dist), taxi geometric distance (Mandist), etc. For the neurons in the field of winning neuron topology, Kohonen rules are used to update, and different distance functions can be used to determine the field, such as Euclidian distance (Dist), taxi geometric distance (Mandist), etc. [28].

Selforgmap, a new self-organizing mapping network function, is selected as the clustering method in this study, which uses the similarity and topological structure of the data itself to cluster the data. Based on the hourly monitoring data of six air quality indexes of 1427 stations in mainland China in 2016, the annual mean and standard deviation of each air quality index of each station were calculated after screening the hourly mean data of 72 dates, and a total of 12 variables were obtained. After Z standardization of the variables, the annual mean and standard deviation of each air quality index were calculated, delete the site data with null data, and use MATLAB software to conduct SOM clustering analysis on variables.

#### 2.2.2. Clustering Analysis Based on SPSS—K-Means Clustering Algorithms

K-means clustering algorithm is an iterative clustering algorithm [29]. The basic idea is to select k points from n sample points as the primary clustering center randomly, then calculate the Euclidean distance from each point of sample to it and assign them to the classes represented by the clustering center with the largest similarity according to the nearest criterion. The clustering center is updated until some termination condition is met. The termination conditions can be that no (or minimum number) objects are reassigned to different clusters, no (or minimum number) clustering center changes again, and the error square and local minimum are similar to SOM clustering steps. In this paper, SPSS software is used to conduct K-means clustering for 12 air quality and fluctuation data.

After the completion of SOM clustering and K-means clustering, according to the classification results, the mean value of each region from high to low is basically consistent with the standard deviation from large to small. This shows that in areas with good air quality, air quality indicators remain at a low level throughout the year, do not fluctuate significantly with dates and seasons, and have a smaller standard deviation. Areas with poor air quality do not suffer from poor air quality all year round, which fluctuate with the date and season due to production, heating and other reasons. Thus, the mean value of the whole year is increased while the standard deviation is larger. Therefore, based on the consistency of mean and standard deviation values of each region, as well as the number of clustering results, this paper divides them into nine categories from high to low.

#### 2.2.3. Spatial Interpolation

The concentration of pollutants is not only affected by local human activities, but also by the mixed diffusion of atmospheric activities. There is a significant spatial correlation between state quantities such as air temperature, that is, the concentration values of pollutants in similar areas are more similar statistically. Due to economic and human resources constraints, the distribution of environmental monitoring sites in China is not uniform, concentrated in the Bohai Rim, Yangtze River Delta and Pearl River Delta and other economically developed regions. Interpolation can understand the complete spatial distribution within the region. The commonly used interpolation methods for regional scale elements include inverse range-weight interpolation and kriging interpolation. The accuracy of the former is greatly affected by the distance to the known points, and it has high requirements on the dispersion and uniformity of the interpolation points, while the results of the unbalanced interpolation points have great fluctuation and poor continuity. Before generating the best estimation method of output surface, the latter should comprehensively calculate the spatial behavior of interpolation point attributes, and the results have good continuity. In this paper, 100 points were randomly selected from 1427 monitoring points in China as verification points, and the cross-validation method was used to verify the interpolation effect. By comparing kriging interpolation and inverse distance weight interpolation, the kriging interpolation with higher accuracy (more than 85%) can scientifically reflect the spatial distribution pattern of air quality and fluctuation in China. Finally, the spatial analysis model of ArcGIS 10.5 is used to verify the spatial autocorrelation of air quality and fluctuation data.

#### 2.2.4. Principal Component Analysis

Principal component analysis (PCA) is a statistical method of dimensionality reduction, which transforms the original random vector with its components dependent into a new random vector with its components unrelated by means of an orthogonal transformation [6]. In this study, four main factors affecting air quality and air quality fluctuation were extracted by PCA and variance maxima rotation.

## 3. Results

### 3.1. The Results of SOM Cluster

In this research, SOM clustering is firstly carried out based on MATLAB. Then, kriging interpolation is used to obtain the clustering distribution of air quality its fluctuation in China, and the spatial autocorrelation is verified. Moran’s I shows that the *p* value is 0, less than 0.05. Z score is 98.43, greater than 1.65, which passed the significance test. The Global Moran index is 0.34, which indicates that China’s air quality and fluctuation is significantly positive spatial autocorrelation (high-value and high-value clusters and low-value and low-value clusters). In this paper, the classification results are firstly classified from high to low according to the values of mean and standard deviation, and nine categories are obtained. Then, the ‘extremely low’, ‘especially low’ and ‘relatively low’ regions are classified as the low-value region. The middle-value region includes the ‘slightly low’, ‘middle’ and ‘slightly high’ regions. Additionally, the ‘relatively high’, ‘especially high’ and ‘extremely high’ regions belong to the high-value region. The spatial distribution of SOM cluster is shown in Figure 3.

Nationally, low-value and middle-value regions are widely distributed and have similar areas, while high-value regions have the smallest and more concentrated areas. Although the high-value regions occupy the lowest proportion in China, its influence cannot be underestimated because they cover a large number of major densely populated areas.

The highest mean region, which is named the ‘extremely high’ region, is divided into three adjacent but discontinuous regions, from northeast to southwest, respectively, the border of Liaoning and Mongolia, central Hebei and the border of Shanxi and Henan. In addition to surrounding the ‘extremely high’ region, the second highest mean region named the ‘especially high’ region is also distributed in central Inner Mongolia and eastern Xinjiang. The lowest mean region named the ‘extremely low’ region is mainly located in the Qinghai-Tibet Plateau and Yunnan-Guizhou Plateau. It is also scattered in the southeast coast, Heilongjiang and northern Xinjiang. Among them, Yunnan, Hainan and Taiwan almost all belong to the ‘extremely low’ region.

Taking the Qinling-Huaihe line as the boundary, all the high-value region and most of the middle-value region are located in the north. The high-value region is concentrated in the east and west, respectively. The eastern concentrated area is mainly in the shape of a strip, covering from the Guanzhong Basin to the north of the Yellow River in Henan to the northeast, and covering most parts of Shanxi and Hebei as well as Beijing and Tianjin, and ending at the border of Liaoning and Mongolia along the Western Liaoning Corridor. There is also a strip distribution in central Inner Mongolia. The western concentrated region is all located in Xinjiang and are roughly divided into two blocky regions, east and west. The middle part of Xinjiang is separated by middle-value and low-value regions, so that the AQFI spatial pattern of Xinjiang is characterized by high values in the east and west and low values in the middle part. Most of the southern region is low-value region. The middle-value region is scattered in the northeast of Sichuan Basin, the middle and lower reaches of Yangtze River and the surrounding area of Hangzhou. Therefore, it can be found that the average value of the Yangtze River basin is higher than that of other regions in the south.

Typical cities were selected according to the principle of spatial uniformity. Baoding, Beijing and Urumqi are selected as the representative cities of the high-value region. The spatial distribution of typical cities is shown in Figure 4. The annual mean values of CO and NO_2_ in Baoding are relatively high, and the mean values of PM_10_ and PM_2.5_ are more prominent. The highest mean value of the former exceeds 150 mg/m^3^, while the mean values of the latter are both around 100 mg/m^3^. At the same time, the standard deviation of each index in Baoding is very large in the case of annual scale, showing a large fluctuation. Standard deviations are generally over 150, with PM_10_ and PM_2.5_ as high as 351 and 262, respectively. This puts Baoding in the ‘extremely high’ region.

The average of all indicators in Beijing is above the average level, with the average PM_10_ around 70 mg/m^3^ and PM_2.5_ around 60 mg/m^3^. There are no outstanding indicators, but it is still at a high level. The standard deviation is large, and each indicator still fluctuates greatly with time. This is similar to the situation in Urumqi, with higher overall indicators. Meanwhile, the standard deviation of PM_2.5_ in Urumqi exceeds 200, and PM_10_ is 285. Therefore, it is classified as the lowest level of the high-value region, that is, the ‘relatively high’ region.

Changchun and Hangzhou are selected as the representative cities of the middle-value region. The mean values of all indicators in Changchun are in the middle level. As a result, it is classified into the highest middle-value region named the ‘slightly high’ region. Most of the indicators in Hangzhou are in the middle, while the average PM_2.5_ level is relatively low compared to other stations, less than 30 mg/m^3^. However, due to the high mean values of NO_2_ and SO_2_, the former is about 20 mg/m^3^, while the latter is over 30 mg/m^3^. Hangzhou is classified as the middle mean of the middle-value region, namely, the ‘middle’ region.

Shenzhen and Guiyang are selected as the representative cities in the low-value region. Except for the relatively high mean value of NO_2_ in Shenzhen, the mean values of other indexes are all at a very low level. The mean of PM_10_ is only 12 mg/m^3^, and that of PM_2.5_ is only 8 mg/m^3^. Therefore, it is classified as the ‘extremely low’ region. All indicators in Guiyang are very low, for example, the mean of PM_10_ is only 9 mg/m^3^, and the mean of PM_2.5_ is only 6 mg/m^3^. In addition, the standard deviation of all indicators throughout the year is at a low level. For example, the standard deviation of SO_2_ and O_3_ is only 21 and 34, respectively. So, Guiyang also belongs to the ‘extremely low’ region.

### 3.2. The Results of K-Means Cluster and Comparison of Two Kinds of Cluster

The results of K-means cluster are obtained by SPSS and shown in Figure 5. We compared the results of K-means clustering and SOM clustering. On the whole, the area size and spatial distribution of the three categories of high, middle and low value in the two results are consistent. However, relatively, the latter’s higher-level areas are larger in size.

SPSS is used to obtain K-means clustering results, and SOM clustering and K-means clustering results are compared (Figure 5). The biggest difference between the two is that the high-value region in K-means clustering results is greatly reduced. The middle-value region remains stable as a whole, but the increase and decrease are different in different regions. The area of the low-value region has increased significantly and occupies the largest proportion in China’s territory.

In terms of grades, among the ‘extremely high’ regions in SOM clustering results, the regional grades of Shanxi-Henan junction, Liaoning-Mongolia junction and Xinjiang have been reduced. Only the central region of Hebei is an ‘extremely high’ region. The high-value region only remains in the position of the ‘extremely high’ region in the original SOM clustering result, and the area was greatly reduced and no longer continuous. The high-value regions in Guanzhong Basin, central Inner Mongolia, central and southern Shanxi, northeastern Hebei, Liaoxi Corridor and Beijing and Tianjin, especially the concentrated regions in Xinjiang, have all been reduced to medium mean values. The middle-value region of K-means clustering is much smaller in northeast and south China, and only exists in the junction of Liaoning and Mongolia, Sichuan and Shaanxi, and southern Anhui. Heilongjiang, Jilin, Chongqing, Hubei, Hunan, Jiangxi and Zhejiang all belong to low-value regions, which have a considerable proportion of middle-value regions in SOM clustering results. However, in northern China, the middle-value region of North China and Xinjiang increased due to the decrease of the high-value region. So, the total area of the middle-value region remains roughly constant. For example, in Xinjiang, the east and west high-value regions have become large areas of middle-value regions. The middle part of the two is separated by a low-value regional corridor. Therefore, Xinjiang still maintains the spatial pattern of high values in eastern and western regions and low values in central regions. As mentioned above, the low-value region mainly shows a trend of substantial increase in northeast and south China. Among them, the area of the ‘extremely low’ region has the most significant growth, covering almost the whole Qinghai-Tibet Plateau and Yunnan-Guizhou Plateau more completely, and accounting for a larger proportion in the southeast coastal provinces.

It can be seen that the spatial distribution of SOM clustering results is more discriminative, while K-means clustering results have a large proportion of low-mean regions. In addition, the high-value region of the former is well consistent with the heavy industry gathering area and the surrounding area of the Gobi Desert in northern China. The middle-value region in the Yangtze River basin also nicely highlights areas that are more industrial than the hills. The ‘extremely low’ regions are distributed in the Altai Mountains, The Greater Khingan Mountains, the Qinghai-Tibet Plateau, the Yunnan-Guizhou Plateau, the mountainous and hilly areas of southeast coastal provinces, as well as Hainan Island and Taiwan Island. This is related to high vegetation coverage, low industrial activity and strong monsoon winds. The results show that the ‘extremely low’ region is in good agreement with climate and topographic region. It is proved that the clustering results and spatial pattern obtained in this paper are objective and accurate.

### 3.3. The Results of AQI Clustering and Comparison with SOM Clustering Results of AQFI

We interpolate the AQI values of each monitoring station and adopt the same classification method as the AQFI for classification (Figure 6). Compared with AQFI, the area of low-value region in the spatial distribution pattern of AQI in China decreased significantly. Additionally, the ‘extremely low’ region is no longer distributed in a large area, showing the characteristics of sporadic small area distribution. The proportion of the ‘relatively low’ region increased to the largest in the low-value region. The middle-value region increases greatly, accounting for the vast majority of China’s land area. The regional grade of Qinghai-Tibet Plateau changes from low mean to medium mean almost. The Yunnan-Guizhou Plateau and southeast coastal provinces also have a large area of low-value region to middle-value region. The location of the high-value region is relatively consistent, and there is no significant deviation, but the area shrinks inward compared with AQFI. The high-value regions in central Inner Mongolia and Guanzhong Basin no longer exist. A large area of high-value region is added in Dabie mountain area, southern Anhui and Qingdao, and scattered in Jiangxi, Guangxi and Guangdong.

Therefore, it can be concluded that the spatial distribution pattern of the AQFI index combined with air quality fluctuations is quite different from the spatial distribution pattern of the AQI index that only considers air quality. The main difference lies in the area proportion of low-value region and middle-value region. In the clustering results of AQFI index, there are more areas with good air quality and small fluctuation range in China.

### 3.4. The Results Principal Component Analysis

Four main factors affecting air quality and air quality fluctuation are extracted by principal component analysis and variance maxima rotation. The composition matrix table of pollutant concentration is shown in Table 2, and the composition matrix of pollutant standard deviation after rotation in Table 3.

The contribution rates of the four main factors affecting air quality are 52.504%, 17.957%, 11.630% and 8.858%, respectively (Table 2), and these four factors can explain 90.949% of all variables. The first principal component is characterized by high positive load on PM_2.5_, PM_10_ and CO concentrations of factor variables. The characteristic of the second principal component is that the factor variable has a high positive load on SO_2_ concentration. The characteristic of the third principal component is that the factor variable has a high positive load on the concentration of NO_2_. The fourth principal component characteristic is that the factor variable has a high positive load on the concentration of O_3_.

The contribution rates of the four main factors affecting air quality fluctuation are 61.733%, 12.379%, 11.028% and 6.956%, respectively (Table 3), and these four factors can explain 92.096% of all variables. The first principal component is characterized by high positive load on PM_2.5_, PM_10_ and CO fluctuations of factor variables. The characteristic of the second principal component is that the factor variable has a high positive load on the fluctuation of SO_2_. The third principal component shows that the factor variable has a high negative load on the fluctuation of O_3_. The fourth principal component characteristic is that the factor variable has a high positive load on the fluctuation of NO_2_.

No matter pollutant concentration or pollutant standard deviation, the first principal component reflects PM_2.5_, PM_10_ and CO concentrations. This demonstrates that PM_2.5_, PM_10_ and CO are the main pollutants affecting air quality and fluctuation.

## 4. Discussion

According to the clustering distribution of air quality and fluctuation, the main conclusion is that the air quality in northern China is worse than that in southern China. Additionally, air quality in northern China fluctuates more sharply. The reasons are analyzed according to the existing theoretical knowledge and can mainly be divided into two aspects: pollutant sources and natural conditions. 

The first is the source of pollutants. Since both the south and the north have automobile exhaust emissions, the difference between the two is mainly concentrated in coal burning [30] and soil dust [31].

In terms of coal burning, due to the uneven spatial distribution of mineral resources and water resources, thermal power generation is the main power supply method in north China [32]. Although a large number of wind turbines have been built in Inner Mongolia, Hebei and other places, due to the unstable wind speed, it is difficult for wind power to be connected to the unified power grid [33]. In the south, thermal power generation is supplemented by a considerable proportion of hydro and nuclear power generation [34,35]. In addition, although there are heavy industry enterprises and factories in both north and south China, the main heavy industry regions in China are still north and northeast China. Air pollutants produced by heavy industry production are significantly higher in the north. At the same time, winter collective heating is basically carried out only in northern China, which consumes a large amount of coal and releases high concentrations of air pollutants such as CO, NO_2_ and SO_2_ into the atmosphere in the process. In addition, the significant periodical heating measures mean that coal burning in the north generates more pollutants in winter than in other seasons. When the longitudinal comparison is made on a yearly basis, the air quality in this area will fluctuate greatly.

From the perspective of soil dust factors, soil dust sources include sand dust and local dust. Because local dust exists in both north and south, the main difference lies in sand dust. The source of dust in northern China lies in the dry climate, low vegetation coverage and low soil and water fixation rate. The desertification grassland in Inner Mongolia Plateau, the Taklimakan Desert in Xinjiang and the Loess Plateau adjacent to the North China Plain are the main sources of very large sandstorms, which have a very significant impact on the air quality in northern China. The fine particles carried by sandstorms can stay high in the air for a long time. This is the leading cause of spring smog in northern China [36]. Although the dust generated in these areas will also spread to the south, due to the effect of distance, the influence in the south will be significantly less than that in the north. In addition, influenced by the relative location of the sandstorm source mentioned above and the cold air moving southward from The Mongolian Plateau in winter, a large amount of the sandstorm is carried to various areas in the north by strong winds. This becomes an important source of pollution in winter. In summer, the southeast monsoon does not bring dust, which is one of the reasons why air quality is more volatile in the north.

Secondly is the natural conditions, the main physical geographical factors affecting air quality are climate [37] and topography [38].

From the perspective of climate factors, specific meteorological conditions will be conducive to reducing the concentration of pollutants in the air, such as rain and snow weather, strong wind and strong convective weather (including hurricane, short-duration heavy rain, convective gusts and thunderstorms). This is also the most important factor affecting local air quality and fluctuations. Not only is the annual precipitation in the north significantly less than that in the south, but the frequency of precipitation is smaller, and the duration of single precipitation is shorter. This makes it harder for pollutants to be washed to the ground by precipitation and still accumulate in the atmosphere. At the same time, when the atmosphere remains calm, the dispersal of pollutants in it slows down and builds up to more severe conditions. Temperature inversion is one of the most typical phenomena. Once this phenomenon is formed, it will make the air unable to convection up and down, and it is difficult for pollutants to diffuse. Compared with the climate conditions in the south, winter begins early in the north, and it is cold at night, with more inversion weather. The high temperature in south China leads to more active vertical atmospheric movement and stronger near-surface turbulence, which is conducive to the diffusion of pollutants. As a result, temperature inversions, which are common in northern winters, cause pollutants to stagnate in the air more frequently, leading to more heavily polluted days in winter and exacerbating annual fluctuations in air quality. In addition, due to the influence of land and sea winds, the diffusion conditions in coastal areas are generally better than those in inland areas. The southern coastline is longer and there are more coastal cities, which is also more conducive to the diffusion of air pollutants.

In terms of topographic factors, the mountainous and hilly terrain in south China makes it easier to form local strong convection, which is more conducive to diffusion than the north China plain in the absence of strong external air currents. At the same time, compared with the north China Plain, the passes and valleys formed between the mountains in the south have a “narrow tube effect” on air flow, which can accelerate the wind speed when the air flows through the region, thus blowing away air pollutants. Based on the above factors, the overall air quality in southern China is kept a high level, and the fluctuation of air quality in the year was basically stable.

This shows that the clustering results and spatial pattern obtained in this paper are objective and explicable.

## 5. Conclusions

In this study, SOM and K-means methods were used to obtain the spatial distribution of China’s air quality and its fluctuation through cluster analysis of the comprehensive index of Air Quality and Fluctuation Index (AQFI), and qualitative analysis of its driving force. The main conclusions are as follows:

Most areas in China are a low-value region with better air quality. The high-value region with poor air quality is the smallest and more concentrated, but it covers more major densely populated areas. Air quality in northern China is worse, and the annual fluctuations of the indicators are more dramatic. All the high-value region and most of the middle-value regions are located in the northern region, which can be divided into east and west concentrated regions. Most of the regions in southern China are low-value regions. It can be found that the value of the Yangtze River Basin is higher than that of other regions in the south.

The Air Quality and Fluctuation Index (AQFI) is an indicator that combines the air pollutant concentration and its standard deviation and has a strong indicative significance for the comprehensive status of air quality and its fluctuation. It can provide better guidance for air treatment in different fields. Compared with the traditional Air quality Index (AQI), AQFI is more comprehensive, and they show different spatial patterns. The main difference lies in the area proportion of the low-value region and middle-value region. In the clustering results of AQFI index, there are more areas with good air quality and small fluctuation range in China.

The spatial distribution of the SOM clustering result is more discriminative, while the K-means clustering result has a larger proportion of low-mean regions. The high-value region of the former is well consistent with the heavy industry gathering area and the surrounding area of the Gobi Desert in northern China. The middle-value region in the Yangtze River basin also highlights areas that are more industrial than the hills. 

According to the results of principal component analysis, PM_2.5_, PM_10_ and CO are the main pollutants affecting air quality and fluctuation, followed by SO_2_, NO_2_ and O_3_.

Since this paper only studies the influence of atmospheric composition changes on air quality in China, without considering the comprehensive effect of pollutants and meteorological elements, it is expected that research in this field will be further strengthened in the future. Due to the large scale of the study area, the characteristics of natural environment and socio-economic differentiation need reasonable consideration and careful demonstration. In addition, due to the complexity of atmospheric variability, more accurate cross-validation methods are needed to analyze the effects of the atmospheric composition on air quality to remove randomness in sample selection.

## Figures and Tables

**Figure 1 ijerph-19-04524-f001:**
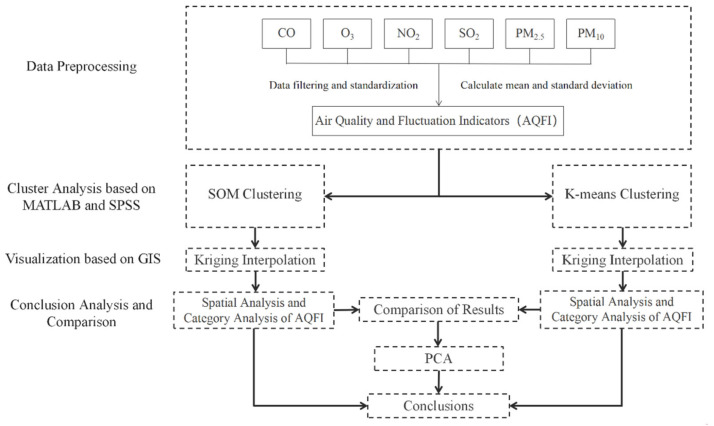
Technology Roadmap.

**Figure 2 ijerph-19-04524-f002:**
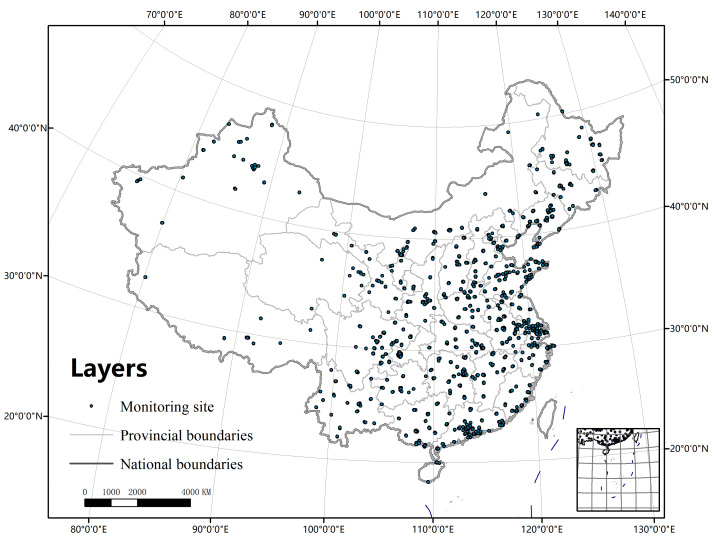
Distribution of meteorological stations.

**Figure 3 ijerph-19-04524-f003:**
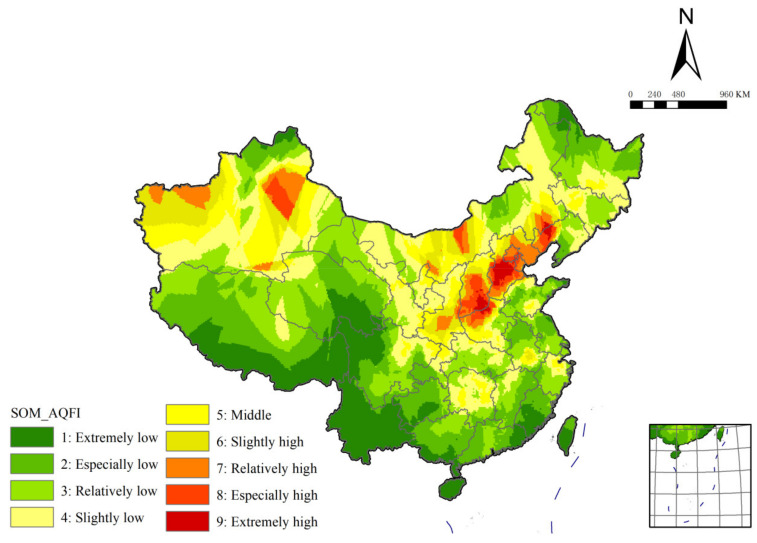
The spatial distribution of SOM cluster.

**Figure 4 ijerph-19-04524-f004:**
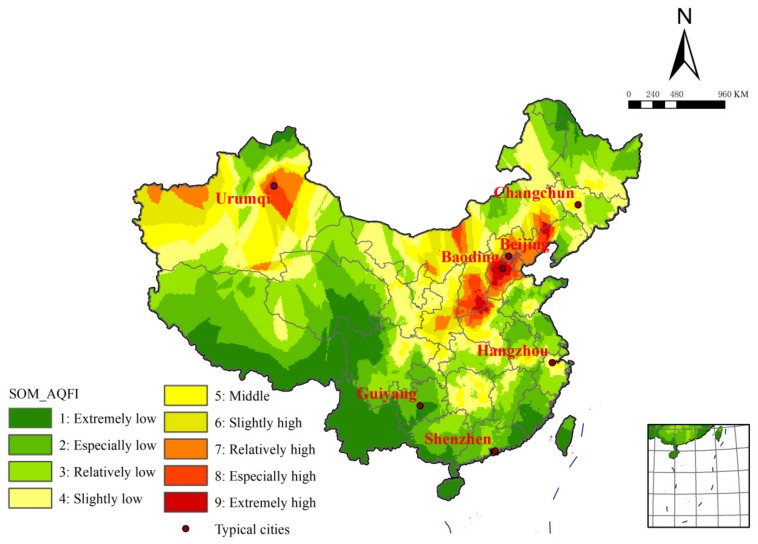
The spatial distribution of typical cities.

**Figure 5 ijerph-19-04524-f005:**
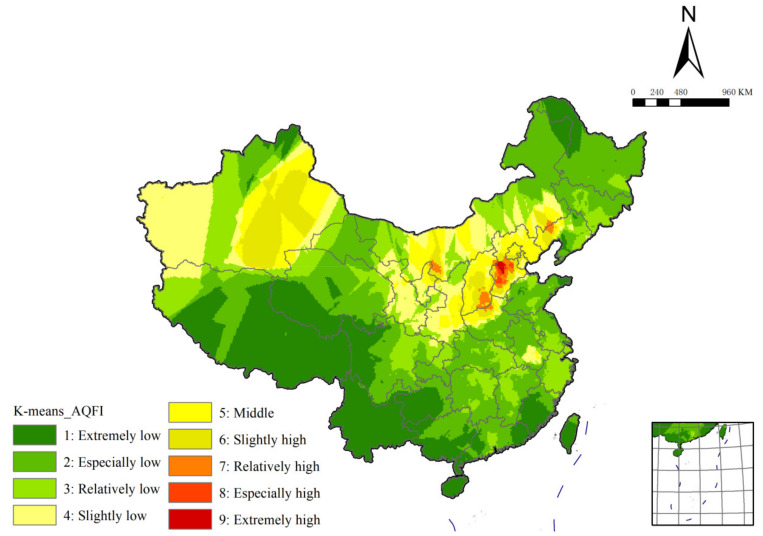
The spatial distribution of K-means cluster.

**Figure 6 ijerph-19-04524-f006:**
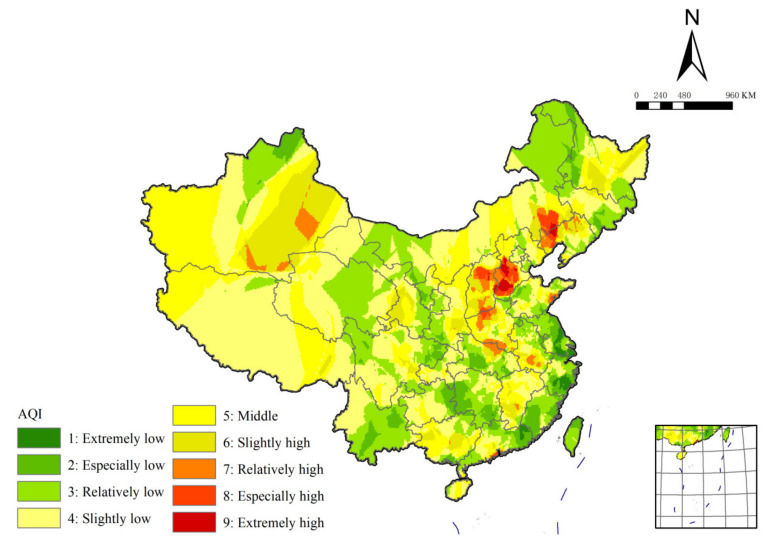
The spatial distribution of AQI.

**Table 1 ijerph-19-04524-t001:** The meanings of air quality indicators.

Indicator	The Meanings of Indicators
CO	Carbon monoxide is one of the main atmospheric chemical pollutants. Due to its high chemical activity, it has a short life span and uneven distribution in the atmosphere. Its variation characteristics basically reflect the source and sink characteristics of the location, and its content has an important influence on the surface environment. Carbon monoxide is toxic and can cause symptoms of different degrees of poisoning at higher concentrations.
O_3_	O_3_ is one of the main atmospheric pollutants in the air, which is the main cause of urban photochemical pollution. O_3_ pollution near the ground causes many hazards to human health, crops and plant growth.
NO_2_	Nitrogen dioxide, toxic, irritating. NO_2_ is mainly formed by fuel combustion and is emitted by cars, trucks, buses, power plants and other sources. It can be emitted directly from combustion sources, but part of it is formed through chemical reactions of nitric oxide and other air pollutants. NO_2_ is an important precursor of anthropogenic ozone and urban smog, and a key factor in the formation of nitric acid, fine particulate matter and nitro polycyclic aromatic hydrocarbons.
SO_2_	Sulfur dioxide, the most common, simplest and irritating sulfur oxide, is one of the major atmospheric pollutants. SO_2_ is a toxic, highly reactive gas with an irritating and putrefying odor that can cause eye and respiratory irritation, bronchoconstriction, cardiovascular disease, cancer, and ecological effects on soil, forests, and fresh water.
PM_10_	PM_10_ is known as atmospheric particulate matter smaller than 10 micrometers in diameter, which has a huge impact on global health. Epidemiological studies have confirmed the long-term and short-term health effects of PM_10_ and further refined the public health effects of PM_10_.
PM_2.5_	PM_2.5_ stands for atmospheric particulate matter with aerodynamic equivalent diameter equal to or less than 2.5 microns, which is the main factor causing haze weather, reducing visibility and affecting traffic safety. PM_2.5_ has become the main pollutant in the air of most cities in China, and PM_2.5_ concentration is an important indicator reflecting the degree of air pollution. Several episodes of severe PM_2.5_ pollution and related problems have aroused widespread concern in society and society.

**Table 2 ijerph-19-04524-t002:** Composition matrix table after rotation of pollutant concentration.

Indicator	Composition				
	PC1	PC2	PC3	PC4	Total
CO	0.76	0.217	0.222	−0.131	/
NO_2_	0.292	0.097	0.947	0.078	/
O_3_	−0.165	−0.063	0.067	0.982	/
PM_10_	0.897	0.19	0.173	−0.098	/
PM_2.5_	0.913	0.141	0.161	−0.11	/
SO_2_	0.276	0.953	0.096	−0.067	/
Variance Contribution (%)	52.504	17.957	11.630	8.858	90.949

**Table 3 ijerph-19-04524-t003:** Composition matrix of pollutant standard deviation after rotation.

Indicator	Composition				
	PC1	PC2	PC3	PC4	Total
CO	0.528	0.535	0.342	0.182	/
NO_2_	0.377	0.218	0.231	0.87	/
O_3_	−0.193	−0.115	−0.947	−0.18	/
PM_10_	0.887	0.246	0.167	0.266	/
PM_2.5_	0.907	0.19	0.164	0.243	/
SO_2_	0.205	0.933	0.072	0.157	/
Variance Contribution (%)	61.733	12.379	11.028	6.956	92.096

## Data Availability

The original data was obtained from the national real-time release platform for urban air quality in China and Beijing Environmental Protection Testing Center. The data of national air quality was available at https://air.cnemc.cn:18007/ (accessed on 17 December 2020). The data of air quality in Beijing was available at http://www.bjmemc.com.cn/ (accessed on 17 December 2020).

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
