# Peer review of "Study on Air Quality and Its Annual Fluctuation in China Based on Cluster Analysis"

_ijerph, 2022, doi:10.3390/ijerph19084524_

Round 1

Reviewer 1 Report

The paper calculates the annual mean and annual standard deviation of seven air quality indicators at 1427 Chinese stations based on hourly-recorded. SOFM, K-means clustering, and PCA were used to process and analyze the collected data.

There are some minor issues that I would encourage the authors to cover for completeness and readability purposes.

  • I would like to see some arguments about the basis on which the selected seven metrics were chosen. Is there a benchmark or references recommending the use of these specific seven metrics?
  • In Tables 2 and 3, I would add the total of the variance contributions to the right of the last row.
  • In Ln 53, it was mentioned that there are six pollutants were selected. Then, in Ln 54, a joining sentence says, " These seven indicators". Please fix or clarify.

Reviewer 2 Report

I’m afraid I cannot support publication of this paper. The air quality index AQI is based on the measurement of particulate matter (PM2.5 and PM10), ozone (O3), nitrogen dioxide (NO2), sulfur dioxide (SO2) and carbon monoxide (CO) emissions. Thus, the authors analysis of both this indicator and at the same time CO, O3, NO2, SO2, PM2.5 and PM10 pollutants, which it consists of, is a wrong assumption. Manuscrypt does not meet the basic requirements for scientific publications, namely it has no novel from the research carried out. The research results and the conclusions presented (points 1-4) are commonly known and do not bring new knowledge to the existing one.

Reviewer 3 Report

Preserving the right quality of air is a very important issue, both for the health of people, their well-being and also general for the future of Earth. Moreover, high air quality promotes the ability to concentrate and thus increases the efficiency of work. Hence, I believe that the subject matter described in the manuscript is very important. The quality of the air can be tested in various ways. The easiest way is to measure air parameters at selected locations, and create the spatial/surface distribution maps. But in such a case, with a large number of measurement points, it will be difficult to perform an effective data analysis due to the possible large differences in measured values between neighboring points, in addition, these differences will depend on the seasons or the occurring weather phenomena. Therefore, proper processing of large amounts of data is necessary to identify the main sources of pollution and to specify remedial actions.

I consider the article important. The authors used a sufficiently large data set. The only thing that raises doubts is the imperfect form of the text, in which there are many repetitions, colloquial and general expressions. In the chapter "disscusion", the authors presented only possible geographic and weather factors that could affect air quality. However, they did not present any specific numerical relationships, for example, to what extent sand in the Tibetan highlands affects the concentration of particulate matter in Chinese cities. In my opinion, it would be enough to check how much Tibetan sand is in this dust. Many of the sentences in the manuscript are general and qualitative only.

I have a few comments to the content of the presented article. The order of the comments does not reflect their significance. It results only from the order of appearance in the text of the manuscript:

  1. Line 58 or 59, “10 meters” – typo, I guess it should be in micrometres.
  2. Line 58 or 59, “concern in society and society” - editing error.
  3. Lines 64-65, “Delete site data with null data after Z normalization of variables.” - incomprehensible sentence.
  4. Line 65, “SPSS” - an explanation of this abbreviation is needed.
  5. Line 66, “SOFM and K-means” - an explanation of these abbreviations is needed.
  6. Lines 74-75, “hourly monitoring data of AQI, CO, O3, NO2, SO2, PM2.5 and PM10 of 1427 stations in China” - once again the same information.
  7. Lines 115-116, “divide the data into K groups, then randomly select K objects as the initial cluster center” - A sentence in this form seems illogical to me. How is it possible to divide the data into k groups and then randomly select K objects from that group; I suspect that just one item has been selected in each group.
  8. Line 221, “meddle” – typo.
  9. Lines 241-242, “But relatively speaking, the area at the higher level of the latter will be wider.” - I suggest rewriting that sentence; technical language is better than colloquial.
  10. Line 333, “strong convection weather” - I believe that there is a strange term, it needs to be clarified.
